# Effects of a Phosphorus-Binding Feed Supplement on the Blood P and Ca Levels in Dairy Cows

**DOI:** 10.3390/ani15070959

**Published:** 2025-03-27

**Authors:** Viktor Jurkovich, Mikolt Bakony, Per Theilgaard, Levente Kovács, Hedvig Fébel

**Affiliations:** 1Centre for Animal Welfare, University of Veterinary Medicine, 1078 Budapest, Hungary; 2Centre for Translational Medicine, Semmelweis University, 1085 Budapest, Hungary; bakony.mikolt@semmelweis.hu; 3ViloFoss, 7000 Fredericia, Denmark; pth@vilofoss.com; 4Institute of Animal Sciences, Hungarian University of Agriculture and Life Sciences, 2100 Gödöllő, Hungary; kovacs.levente@uni-mate.hu; 5Department of Obstetrics and Food Animal Medicine Clinic, University of Veterinary Medicine, 1078 Budapest, Hungary; febel.hedvig@univet.hu

**Keywords:** aluminum sulfate, dairy cows, ionized calcium, phosphorus binding, subclinical hypocalcemia, total calcium

## Abstract

Dairy cows often experience low blood calcium levels after calving, leading to health problems, reduced milk production, and fertility issues. This study tested whether adding aluminum sulfate to the diet of pregnant cows in the final weeks before calving could help prevent this issue. Aluminum sulfate binds phosphorus, a mineral that can interfere with calcium levels. Researchers divided 34 cows into two groups: one received aluminum sulfate, and the other did not. Blood samples were taken before and after birth to measure calcium, phosphorus, and other important indices. Cows that received aluminum sulfate had higher calcium levels in their blood after calving and lower phosphorus levels than cows that did not receive the supplement. Although milk production was slightly higher in the supplemented group, other health indicators were similar between the groups. These findings suggest that aluminum sulfate could be a valuable dietary addition to prevent calcium deficiency in dairy cows.

## 1. Introduction

Hypocalcemia, when the calcium concentration in the blood is lower than the physiological level, is a common metabolic disorder in dairy cows in the transition period. Its more frequent form is subclinical hypocalcemia (SCH) [1], which causes no clinical signs, but makes the freshly calved cow prone to metritis, mastitis, subclinical, or clinical ketosis, and impaired rumen motility, thus having a negative effect on milk production and reproduction [2,3,4,5]. In the more severe form, milk fever or clinical hypocalcemia (CH) can lead to weakness or even death [1]. Hypocalcemia is a frequent problem in dairies worldwide, as about 29–64% of multiparous and 5–25% of primiparous cows experience SCH [1,6,7,8], while CH occurs in 2.2–4.5% of cows in the days after calving [1,7].

The diagnosis of milk fever is based on its clinical appearance or blood concentrations of total Ca (tCa) or ionized Ca (iCa, the active form of Ca in metabolic processes). At the same time, to diagnose SCH, blood Ca levels must be tested. Several thresholds for diagnosing SCH are reported in the literature, as reviewed by Couto Serrenho et al. [1], and the most commonly used thresholds are tCa concentrations below 2.1 mmol/L or iCa concentrations below 1.05 mmol/L [9]. Ionized Ca concentration is considered the more reliable parameter for diagnosing SCH [10,11].

Hypocalcemia occurs most frequently during the onset of lactation when the Ca mobilization capacities of the animal body cannot fulfill the high Ca demands for milk yield. Decreasing blood Ca concentration after parturition activates the parathyroid hormone (PTH), which increases Ca release from the bones, Ca absorption from the intestinal tract, and the renal reabsorption of Ca [12,13]. However, the Ca demands of high-yielding dairy cows may increase more rapidly than the pace of mobilization, which can lead to a deficiency in the available Ca. Magnesium is involved in Ca homeostasis, as low concentrations of serum Mg attenuate PTH release in response to low serum Ca, and low serum Mg results in vitamin D and PTH insensitivity [12,13].

There are several strategies for the prevention of hypocalcemia after parturition. The most straightforward is oral calcium supplementation after calving [12,13]. However, that is not the most commonly used method. Limiting Ca uptake before calving is a more frequently used nutritional method to prevent hypocalcemia. The parathyroid hormone becomes activated during increased Ca demand if transient hypocalcemia is induced in the period preceding calving [14]. Reducing the dietary cation–anion difference (DCAD) is a common nutritional strategy. A negative DCAD disrupts the electroneutrality of bodily fluids and slightly shifts the acid-base balance. It induces the Ca excretion that activates the parathyroid hormone and provides favorable conditions for Ca to be present in the circulation in its free (ionized) form [12,13].

More recently, the reduction or binding of the phosphorus content in the diet has also been effective in preventing hypocalcemia [12,13,15,16,17]. Aluminum compounds, primarily silicate, have already been used as feed additives to prevent hypocalcemia [17,18,19,20]. Previously, it was thought that Al-silicate binds Ca in the diet and thus reduces Ca absorption, thereby acting in the same way as a low-calcium diet. However, research showed that the plasma phosphorus concentration was significantly reduced in cows fed with Al-silicate [21]. Indeed, phosphorus supplementation in feed could reduce the effects of Al-silicate on hypocalcemia [22]. Therefore, Al-silicate may improve plasma Ca levels by reducing the available dietary phosphorus.

There is no information on the effects of other aluminum compounds in preventing SCH in dairy cows. Aluminum sulfate (E 520), aluminum sodium sulfate (E 521), aluminum potassium sulfate (E 522), aluminum ammonium sulfate (E 523), sodium aluminum silicate (E 554), and potassium aluminum silicate (E 555) are approved food additives [23]. In the US, the FDA lists aluminum sulfate as GRAS (Generally Recognized as Safe without Limits [24]). It was hypothesized that aluminum sulfate might effectively substitute aluminum silicate in the dairy ration. This study aimed to test the effects of supplementing Al sulfate on the close-up ratio of Holstein cows. We hypothesized that Al sulfate reduces the plasma phosphorus concentration and promotes optimal Ca availability in early postpartum dairy cows. We also aimed to measure the Al concentration of the milk produced after approximately three weeks of aluminum exposure to investigate food safety.

## 2. Materials and Methods

### 2.1. Animals, Husbandry, Feeding Regime

All the experimental procedures applied in this study were reviewed and approved by the Government Office in Pest County, Budapest, Hungary (permit No: PE/EA/1076-5/2020).

The study was conducted on a large-scale commercial dairy farm in Western Hungary (Ják-Felsőnyírvár, 47°07′04.6″ N 16°34′07.4″ E), housing around 600 cows and their offspring. The study was performed between February and April 2022. Dry cows were chosen for the study and paired according to parity, 305 d milk production in the previous lactation, the length of the previous lactation, and the periparturient anamnesis of the last post-calving period (incl. milk fever, ketosis, metritis, or displaced abomasum cases; Table 1). In total, 34 cows were assigned to the Control (CTRL, *n* = 17) and Treatment (TRT, *n* = 17) groups, respectively. The cows were moved to the close-up group at least 14 days before the expected calving date. Both groups were housed in straw-bedded barns with an ad libitum water supply before calving. They received the same low-Ca total mixed ratio (TMR) without using anionic salts or other compounds that would influence Ca balance (Table 2 and Table 3). The daily amount of TMR in both groups was calculated to allow for a 5% refusal rate, ensuring ad libitum feeding. The cows of the TRT group were given Al sulfate at a dose of 400 g/day/cow until calving. As the TMR was offered ad libitum, sufficiently to allow for at least 5% refusal, the measured amount of Al sulfate in the diet feeder was increased by 5%. The TMR was delivered once a day (between 08:00 and 09:00 h). Feed was pushed up three times per day. After calving, the cows were moved to the fresh milking group and housed in a free-stall barn with straw bedding and an ad libitum water supply. The composition of the TMR changed adaptively (Table 2 and Table 3). The ration was adjusted daily to achieve a minimum 5% refusal and fed in two deliveries (07:00 and 17:00 h). The TMR offered did not differ between the CTRL and TRT groups. The cows were milked twice daily (06:00 and 16:00 h) in a parallel milking parlor. The body condition was scored at enrollment and 14 days postpartum.

### 2.2. Samplings and Laboratory Analysis

Blood samples were taken on 13 occasions between enrolment and 14 days postpartum: 14, 10, 7, 5, 3, and 1 day(s) before expected calving, 6–12 h after calving, and also 1, 2, 3, 5, 7, and 14 days postpartum.

Blood samples were taken from the caudal vessels between 08:00 and 10:00 a.m. and sorted into sampling tubes without anticoagulant (Vacuette, Greiner Bio-One International GmbH, Kremsmünster, Austria) and into tubes containing Li-heparin (Vacuette, Greiner Bio-One International GmbH, Kremsmünster, Austria). Blood collected in Li-heparin tubes was used to measure iCa. Serum samples were stored at 4 °C after sampling, centrifuged within 6 h, and then stored at −20 °C until laboratory analysis.

Ionized Ca was measured in whole blood samples with the LAQUAtwin Ca-11 blood iCa measuring device (HORIBA Advanced Techno. Co., Ltd., Kyoto, Japan; [25]). From the serum samples, the tCa, P, Mg, and beta-hydroxybutyrate (BHB) concentrations were measured using the Siemens Advia 1800 Chemistry System (Siemens Healthcare Diagnostics Inc., Tarrytown, NY, USA). The serum concentrations of the studied minerals were determined using the following methods: tCa: the arsenazo method [26]; P: the phosphomolybdate method [27]; and Mg: the xylidyl blue complex method [28]. The total Ca, Mg, and P were determined with reagent kits No. 43941, No. 41341, and No. 41341, produced by Diagnosticum Zrt. (Budapest, Hungary), respectively. The serum BHB concentration was determined using a reagent kit (BHB 21 FS) manufactured by DiaSys GmbH (Holzheim, Germany), according to the method outlined by Li et al. [29].

Individual milk samples were collected on days 2 and 7 postpartum, just before the morning milking. The teats were disinfected with 70% alcohol before sampling. Then, after checking the first milk drops, an equal amount of milk from each teat, 10 mL per animal, was sampled in a sterile tube. After sampling, the samples were cooled to 4 °C and stored at −20 °C until analysis. The aluminum concentrations were determined from the milk samples by an inductively coupled plasma mass spectrometer (Agilent 7700X ICP-MS; Agilent Technologies, Santa Clara, CA, USA).

Data on the daily milk yield of the cows were collected from the farm database and averaged to obtain weekly milk yields for statistical analysis.

### 2.3. Statistical Analysis

The baseline parameters were compared using Fisher’s exact test and the Welch test. The blood parameters and milk yield means were also compared between the CTRL and TRT groups at different sampling times. Repeated measurements on the same animal were considered using linear mixed models with ‘group’, ‘sampling time’, and their interaction term as predictors and ‘cow ID’ as the random term. Model fit was evaluated using diagnostic plots. In the case of a non-normally distributed variable (BHB), logarithmic transformation was applied before model fitting to achieve the normality of residuals. The proportion of cows diagnosed with hypocalcemia at different sampling times was compared with Fisher’s exact test. The number of hypocalcemia days (both iCa and tCa based) was compared using the Wilcoxon–Mann–Whitney test. The minimum iCa and total Ca concentrations were compared using the Welch test. Comparisons were made between the control and treatment groups for the total length of the study, the prepartum and postpartum periods, respectively.

All statistical analyses were performed in the R statistical environment [30]. The packages lme4, lmerTest, and emmeans were used to fit the models and perform multiple comparisons between groups at different sampling times. Statistical significance was set at *p* < 0.05.

## 3. Results

There were no baseline differences between the experimental and control groups regarding the basic production parameters and body condition scores at the start of the study period. The average body condition scores did not differ at the end of the observation either (2.97 ± 0.37 and 3.00 ± 0.39 in CTRL and TRT groups, respectively, *p* = 0.8250). The cows spent at least 14 days in the pre-calving group (CTRL: 21.5 ± 9.5; TRT: 18.6 ± 7.1). The first sampling at enrolment is indicated as −14 d prepartum in the tables and figures.

The total Ca levels of the CTRL and TRT groups differed only at 6–12 h postpartum and on days 1 and 2 after calving. On average, the serum tCa concentration of TRT cows was 0.22, 0.18, and 0.14 mmol/L higher than that of CTRL cows. There was no difference in the tCa at the other sampling times (Figure 1, Table 4). The ionized Ca levels in the blood of TRT cows were higher from day 10 before calving to day 3 after calving (Figure 2, Table 4). After that, the iCa did not differ between the two groups.

The treatment was associated with lower P concentrations in the TRT group compared to the CTRL group during the Al feeding period (before calving). This value remained lower after calving until day 2 (Figure 3, Table 4); thereafter, there was no difference between the groups. The serum Mg concentrations were higher in the CTRL group compared to the TRT group at day 5 before and 12 h after calving, and there was no difference at the other time points (Figure 4, Table 4). The serum BHB levels increased in the CTRL group after calving and reached a peak at on day 7, exceeding the threshold for subclinical ketosis. BHB in the TRT group remained under the subclinical ketosis threshold. However, the BHB levels differed between groups only 14 days after calving, with concentrations on average 1.42 (95% CI: 1.01; 1.98) times higher in CTRL cows than in TRT ones (Figure 5 and Table 4).

The TRT group had a lower occurrence of subclinical hypocalcemia (tCa < 2.1 mmol/L or iCa < 1.05 mmol/L). In the CTRL group, approximately 60% of the animals were diagnosed with SCH 12 h after calving, which is more than twice that of the TRT group (24%). At 24 and 48 h after calving, the incidence of SCH continued to be higher in the CTRL group than in the TRT group (Table 5). Table 6 displays the number of hypocalcemia days and the minimum ionized and total Ca concentrations for each group during the total study period, as well as the prepartum and postpartum periods.

At week 3 postpartum, the daily milk yield tended to be higher in the TRT group (Figure 6, Table 7) than in the CTRL group.

The aluminum concentration in milk was very low in both the day 2 and day 7 samples (Table 8). In most samples (TRT day 2: 13/17; day 7: 12/16; CTRL: day 2: 11/16; day 7: 13/17), the aluminum concentration was under the detection limit (0.1 mg/L) of the ICP-MS method. In the remaining samples, the Al concentration was similar in the two groups.

## 4. Discussion

To our knowledge, this is the first study to investigate the effects of the prepartum supplementation of aluminum sulfate on Ca, P, and Mg homeostasis in dairy cows. Using aluminum sulfate in dairy cows was expected to be similar to using sodium aluminum silicate (Zeolite A), as both contain aluminum. Several studies measured the effects of supplemental sodium aluminum silicate during the close-up period on dairy cows [17,18,19,20,21,22,31,32,33,34].

In all but one of the studies above, the plasma P concentrations in Al-sulfate-supplemented cows were lower compared to the control. The study by Khachlouf et al. [33] was an exception, in which 200 g of sodium aluminum silicate was given daily to pre-calving cows for 40 days, while the blood P level did not decrease. In the studies cited above, blood P levels usually fell below the lower limit of the reference range (1.3 mmol/L; [35]) during aluminum supplementation, but returned to normal levels within the first week after calving.

Similarly, the serum P levels in our study were lower in the TRT group compared to the CTRL group, with values below the lower reference threshold. Non-supplemented cows also showed a slight hypophosphatemia after calving. Serum P levels returned to normal 5 days after calving in both groups. As previously mentioned, several studies have linked the lower plasma P concentrations to a reduced dietary P availability when feeding sodium aluminum silicate. The partial degradation of sodium aluminum silicate is thought to release Al, which forms insoluble phosphate complexes in the intestinal lumen [31]. If the phosphorus concentration in the plasma is higher than 2.0 mmol/L, it can inhibit the conversion of 1,25-dihydroxy vitamin D from 25-hydroxyvitamin D. Without this conversion, Ca absorption activity cannot be increased in the small intestine [12,36]. A mild degree of hypophosphatemia is therefore necessary to initiate Ca mobilization.

Indeed, feeding diets with reduced P concentrations benefited the blood Ca concentrations in dry cows during the last 4 weeks of pregnancy [37]. Cohrs et al. [15] observed that hypocalcemia was less frequent among cows fed with low P diets (0.15% pre-calving and 0.20% post calving) compared to cows fed with adequate P diets (0.28% pre-calving and 0.44% post calving). They concluded that Ca mobilization from bone was stimulated by the PTH and additional factors associated with low P levels in the blood [15]. Köhler et al. [38] found lower FGF23 expression in non-lactating, non-pregnant sheep consuming low-P but normal-Ca-level feed, and the expression of several other genes related to vitamin D metabolism was also altered. Feeding dairy calves 0.2% dietary Al for 5 days at 7 weeks increased P excretion in feces but not in urine; thus, P absorption was significantly reduced [39]. In another study, dietary Al supplementation decreased serum P concentrations and the apparent absorption of Ca and increased urinary Ca excretion in lactating beef cows [40]. This indicates that Al-containing feed may have increased the PTH levels. Our results are similar to the studies discussed; blood iCa and serum tCa concentrations were significantly higher in the TRT group. The serum iCa levels showed a more significant difference between the TRT and CTRL groups than the tCa levels, suggesting that the iCa concentration was more sensitive to changes in Ca homeostasis. Others have confirmed that iCa measurement is more precise in diagnosing hypocalcemia than tCa [10,11]. Aluminum supplementation reduced the occurrence of subclinical hypocalcemia in our study.

Several studies have shown decrease in blood Mg concentrations during the pre-calving period with sodium aluminum silicate supplementation [19,20,31,32,34]. However, other studies found no significant effect of sodium aluminum silicate supplementation on the serum Mg levels [21,22,33]. In several cases, the serum Mg concentration remained within the reference range of 0.75–1.0 mmol/L [36], even if the concentration in the treated cows decreased compared to the control animals [20,41]. The exact mechanisms underlying the observed decrease in serum Mg concentration remain unclear. Still, sodium aluminum silicate is assumed to bind Mg directly, reducing the Mg availability for the cow [14,21].

In our study, the magnesium levels were within the normal range in both groups, with a slight decrease in Mg after calving (Figure 4). According to others [22,31,41], serum Mg levels are less well-regulated than Ca levels in pre-calving cows, and there is essentially a balance between absorption by the rumen and intestines and excretion via the kidney. Consequently, a regular diet without supplementation in the pre-calving period can control the Mg levels in well-managed herds. Our data show that, although a 14–21 d prepartum supplementation with a dose of 400 g of Al sulfate/cow/day may result in minor changes in the serum Mg concentration, this is only temporary, without any harmful effects on health.

Subclinical ketosis is highly frequent in dairy cows after calving [42,43,44], with the highest prevalence between 1 and 10 days postpartum [43], which was also found in our study. BHB showed a pattern of increasing after calving until day 7 and then decreasing until day 14. The BHB values exceeded the threshold set for the diagnosis on day 7 of subclinical ketosis in the control group, but remained below the threshold in the treatment group. Calcium is necessary for muscle function, including muscles in the rumen wall, and is thus crucial in rumen motility and rumination activity [35]. Disrupted Ca metabolism is associated with decreased feed intake and subclinical ketosis [1]. Balanced Ca homeostasis may thus result in better rumen motility and increased feed intake. We measured slightly lower BHB levels in the TRT group at all sampling times, with a statistically significant difference observed at day 14 postpartum, which may be attributed to improved rumination activity and increased feed intake. Measuring individual feed intake was impossible on the farm; however, cows consumed the whole amount of the daily TMR. Several other studies have measured the effects of aluminum silicate on the energy metabolism and balance of dairy cows, primarily examining serum BHB levels. Some showed significantly lower BHB levels in the treated group [20], while others found no difference between the control and the treated groups [33,34,45].

In our study, the milk yield did not differ between the two groups. There was only a tendency at week 3 in favor of the experimental group. Some studies found an increased milk yield using aluminum silicate supplementation [33]. However, others found no difference in milk yield between the treated and control groups [20,21,31,32,34].

Given the cumulative nature of aluminum in the organism after dietary exposure, the EFSA considered it more appropriate to establish a tolerable weekly intake (TWI) rather than a tolerable daily intake such as in human nutrition [23]. The EFSA established a TWI of 1 mg Al/kg bw/week. The estimated daily dietary exposure to Al in the general population, assessed in several European countries, varied from 0.2 to 1.5 mg/kg bw/week at the mean and was up to 2.3 mg/kg bw/week in highly exposed consumers [23]. The Al milk concentration in our study was under the detection limit in most samples (<0.1 mg/L). The Al levels that could be measured in the milk samples were similar to what Frizzarini et al. [45] found in their study on the effects of aluminum silicate, and similar to those of others studies (range: 0–0.3 mg/L; [46,47,48]) measuring microelements in milk. Overall, the Al concentration was negligible in the milk samples in our study compared to other food products [47,49], indicating no difference between the CTRL and TRT groups. These findings align with the statements of Boudebbouz et al. [50], namely that humans’ exposure to Al through cow milk consumption is considered safe.

## 5. Conclusions

In conclusion, 400 g/cow/day dose of Al sulfate applied in the close-up diet reduced the plasma P levels and improved the serum tCa (at 12 h and on day 1 and day 2 after calving) and iCa (throughout the experimental period until day 2 postpartum) concentrations. Therefore, it is suitable for preventing subclinical hypocalcemia in dairy cows. Also, the prepartum supplementation of 400 g/day Al sulfate for dairy cows can be considered appropriate regarding food safety.

## Figures and Tables

**Figure 1 animals-15-00959-f001:**
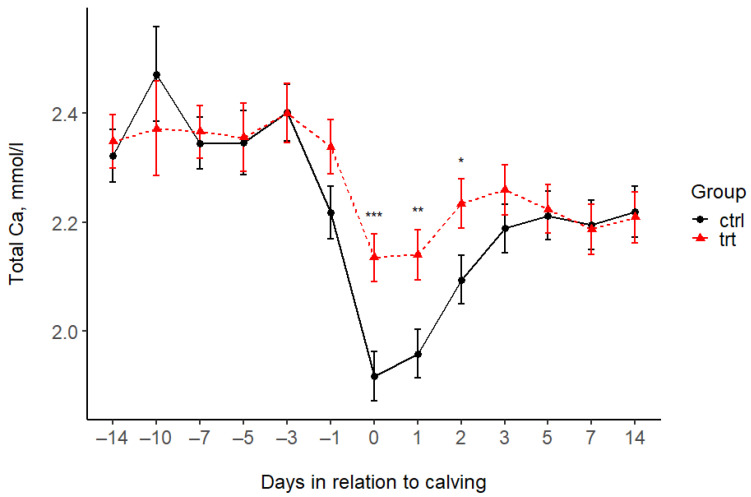
Serum total calcium concentrations (mean ± SE) in the treated (TRT) and control (CTRL) groups during the study period. −14 = sampling at enrolment. 0 = 12 h after calving. Asterisks indicate statistically significant differences: * *p* < 0.05, ** *p* < 0.01, and *** *p* < 0.001.

**Figure 2 animals-15-00959-f002:**
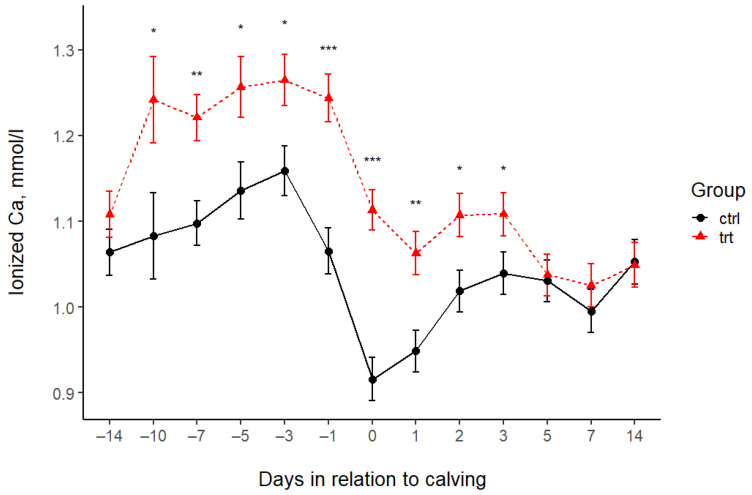
Blood ionized calcium concentrations (mean ± SE) in the treated (TRT) and control (CTRL) groups during the study period. −14 = sampling at enrolment. 0 = 12 h after calving. Asterisks indicate statistically significant differences: * *p* < 0.05, ** *p* < 0.01, and *** *p* < 0.001.

**Figure 3 animals-15-00959-f003:**
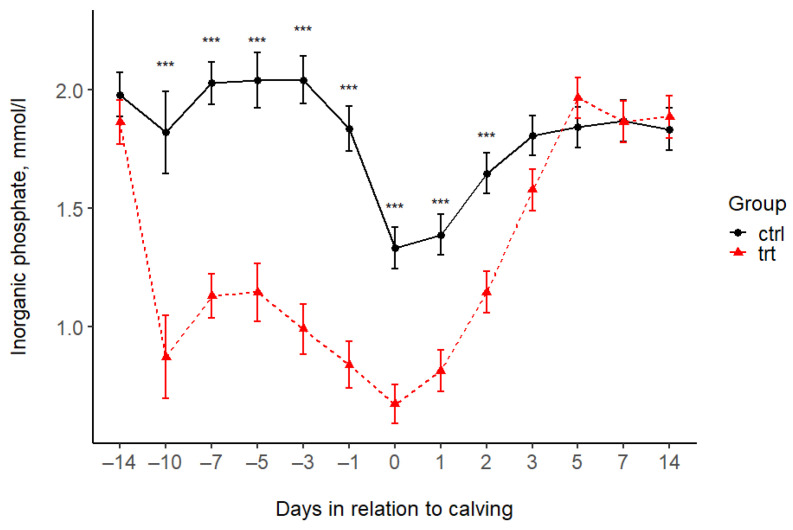
Serum phosphorus concentrations (mean ± SE) in the treated (TRT) and control (CTRL) groups during the study period. −14 = sampling at enrolment. 0 = 12 h after calving. Asterisks indicate statistically significant differences: *** *p* < 0.001.

**Figure 4 animals-15-00959-f004:**
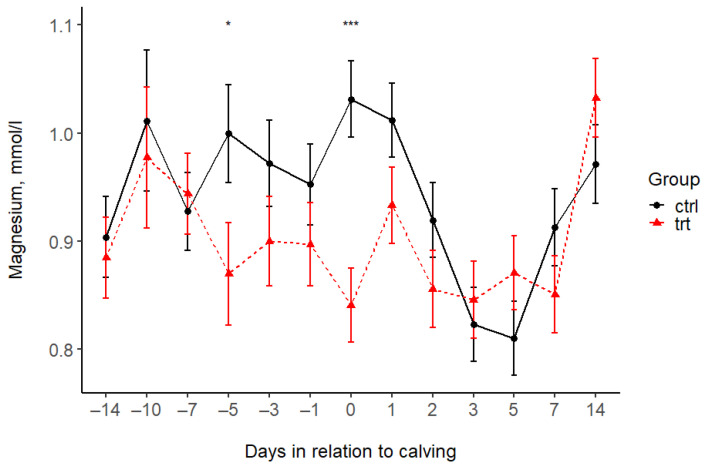
Serum magnesium concentrations (mean ± SE) in the treated (TRT) and control (CTRL) groups during the study period. −14 = sampling at enrolment. 0 = 12 h after calving. Asterisks indicate statistically significant differences: * *p* < 0.05, and *** *p* < 0.001.

**Figure 5 animals-15-00959-f005:**
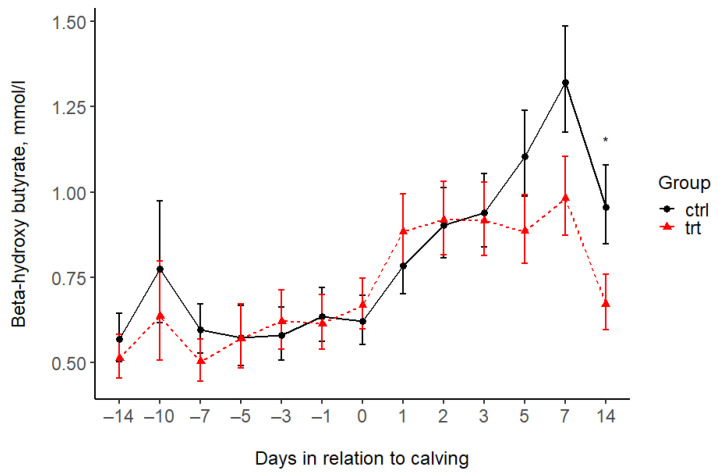
Serum beta-hydroxybutyrate concentrations (mean ± SE) in the treated (TRT) and control (CTRL) groups during the study period. −14 = sampling at enrolment. 0 = 12 h after calving. Asterisks indicate statistically significant differences: * *p* < 0.05.

**Figure 6 animals-15-00959-f006:**
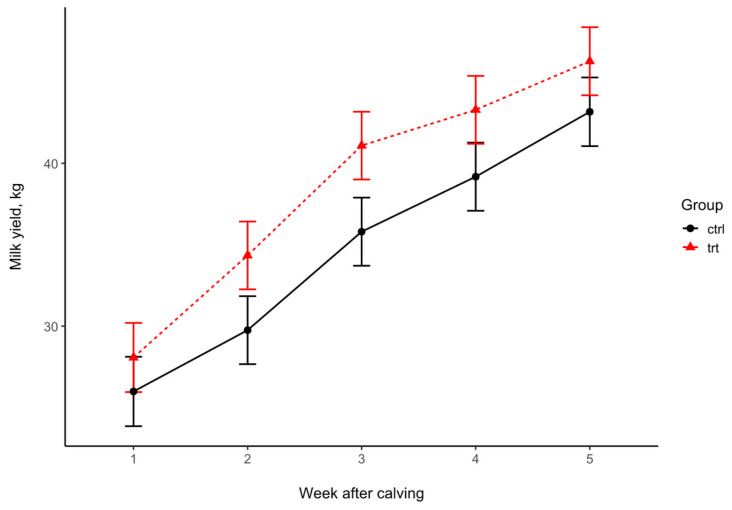
Weekly average milk yield (mean ± SE) in the treated (TRT) and control (CTRL) groups during the first 30 days of lactation.

**Table 1 animals-15-00959-t001:** Data of the cows enrolled in the study.

	Control	Treated	*p*-Value
Parity (±SD)	2.9 ± 1.1	3.2 ± 1.3	1.000
Second lactation (animal/group)	8	7	
Third lactation	3	3	
Fourth lactation	5	4	
Fifth lactation	1	2	
Sixth lactation	0	1	
Milk production for 305 days in the previous lactation (kg ± SD)	11,241 ± 2544	11,318 ± 1962	0.9199
Length of the previous lactation (days ± SD)	344 ± 116	351 ± 86	0.9109
Number of metritis cases in the previous lactation	0	1	
Number of displaced abomasum cases in the previous lactation	0	0	
Number of milk fever cases in the previous lactation	0	0	
Number of ketosis cases in the previous lactation	1	0	
Body condition score (mean + SE)	3.62 ± 0.46	3.53 ± 0.37	0.5394

**Table 2 animals-15-00959-t002:** Ingredients of the TMR before and after calving.

Ingredients (kg)	Before Calving		After Calving
	Control	Treated	
Corn silage	14.0	14.0	17.0
Wheat straw	4.0	4.0	0.0
Water	4.0	4.0	4.0
Concentrate	5.5	5.5	11.0
Barley haylage	0.0	0.0	7.0
Wet corn gluten feed	0.0	0.0	5.0
Meadow hay	0.0	0.0	1.0
Aluminum sulfate	0.0	0.4	0.0
In total	27.5	27.9	45.6

**Table 3 animals-15-00959-t003:** Calculated chemical composition of the TMR before and after calving.

Items	Before Calving		After Calving
	Control	Treated	
Dry matter (%)	47.8	47.8	48.3
NE_l_ (MJ/kgDM)	5.9	5.9	7.5
Crude protein (%DM)	13.7	13.7	19.7
Rumen degradable protein (%DM)	9.3	9.3	13.5
Rumen undegradable protein (%DM)	4.2	4.2	6.2
Sugar (%DM)	2.9	2.9	4.9
Starch (%DM)	20.9	20.9	23.7
Crude fiber (%DM)	22.3	22.3	13.5
ADF (%DM)	25.4	25.4	15.4
NDF (%DM)	39.9	39.9	30.4
NFC (%DM)	35.7	35.7	39.2
Ether extract (%DM)	3.2	3.2	4.5
Ash (%DM)	7.3	7.3	7.1
Ca (%DM)	0.59	0.59	0.89
P (%DM)	0.42	0.42	0.43
Ca:P ratio	1.41	1.41	2.07
Mg (%DM)	0.25	0.25	0.39
Cl (%DM)	0.44	0.44	0.32
S (%DM)	0.20	1.08	0.28
DCAD (mEq/100 g DM)	10.6	−13.2	29.1

**Table 4 animals-15-00959-t004:** The measured biochemical parameters in the treated and control groups during the study period.

	Days in Relation to Calving														*p*-Value		
	−14 *	−10	−7	−5	−3	−1	0 **	1	2	3	5	7	14	SEM	Group	Sampling	G × S
Total Ca (mmol/L)																	
Control	2.32	2.47	2.34	2.35	2.40	2.22	1.92	1.96	2.09	2.19	2.21	2.19	2.22	0.05	0.134	<0.001	0.043
Treated	2.35	2.37	2.37	2.35	2.40	2.34	2.14	2.14	2.23	2.26	2.22	2.19	2.21	0.05			
*p*-value (pairwise)	0.702	0.415	0.757	0.913	0.986	0.085	0.001	0.005	0.028	0.268	0.852	0.903	0.874				
Ionized Ca (mmol/L)																	
Control	1.06	1.08	1.10	1.14	1.16	1.07	0.92	0.95	1.02	1.04	1.03	1.00	1.05	0.03	<0.001	<0.001	0.0005
Treated	1.11	1.24	1.22	1.26	1.26	1.24	1.11	1.06	1.11	1.11	1.04	1.03	1.05	0.03			
*p*-value (pairwise)	0.246	0.025	0.001	0.013	0.012	<0.001	<0.001	0.001	0.012	0.049	0.838	0.402	0.913	0.05			
Phosphorus (mmol/L)																	
Control	1.98	1.82	2.03	2.04	2.04	1.84	1.33	1.39	1.65	1.81	1.84	1.87	1.83	0.1	<0.001	<0.001	<0.001
Treated	1.86	0.87	1.13	1.14	0.99	0.84	0.67	0.81	1.15	1.58	1.96	1.86	1.88	0.1			
*p*-value (pairwise)	0.378	<0.001	<0.001	<0.001	<0.001	<0.001	<0.001	<0.001	<0.001	0.061	0.305	0.984	0.686				
Magnesium (mmol/L)																	
Control	0.90	1.01	0.93	1.00	0.97	0.95	1.03	1.01	0.92	0.82	0.81	0.91	0.97	0.04	0.152	<0.001	0.002
Treated	0.88	0.98	0.94	0.87	0.90	0.90	0.84	0.93	0.86	0.85	0.87	0.85	1.03	0.04			
*p*-value (pairwise)	0.719	0.713	0.755	0.048	0.209	0.302	<0.001	0.112	0.197	0.645	0.214	0.215	0.233				
BHB (mmol/L) ***																	
Control	0.57	0.77	0.60	0.57	0.58	0.64	0.62	0.78	0.90	0.94	1.11	1.32	0.96	0.07	0.259	<0.001	0.568
Treated	0.51	0.64	0.50	0.57	0.62	0.61	0.67	0.88	0.92	0.92	0.89	0.98	0.67	0.07			
*p*-value (pairwise)	0.565	0.539	0.330	0.983	0.728	0.848	0.648	0.466	0.923	0.871	0.167	0.072	0.038				

* −14 = sampling at enrolment. ** 0 = 12 h after calving. *** The displayed means and standard errors are retransformed from the log scale, and the *p*-values are from the log (BHB) model. Note: the parameters of the log model become multipliers when retransformed to the original scale.

**Table 5 animals-15-00959-t005:** The number and percentages of subclinical hypocalcemia cases defined as having a plasma total calcium concentration below 2.1 mmol/L or ionized Ca concentration below 1.05 mmol/L in the treated and control groups.

	Subclinical Hypocalcemia (tCa < 2.1 mmol/L)			Subclinical Hypocalcemia (iCa < 1.05 mmol/L)		
Time After Calving	Control	Treated	*p*-Value	Control	Treated	*p*-Value
12 h	10 (59%)	4 (24%)	0.0798	12 (71%)	5 (29%)	0.0380
24 h	7 (41%)	3 (18%)	0.2587	13 (77%)	6 (35%)	0.0366
48 h	5 (29%)	3 (18%)	0.6880	11 (65%)	5 (29%)	0.0844

**Table 6 animals-15-00959-t006:** The number of hypocalcemia days and the minimum ionized and total Ca concentrations between groups for the total study period, prepartum and postpartum periods, respectively.

Item	Period	TRT	CTRL	*p*-Value
The number of days in hypocalcemia, based on iCa concentrations	total	3(2–4)	5(4–7)	0.0015
	prepartum	0(0–0)	1(0–2)	0.0082
	postpartum	3(2–3)	4(3–5)	0.0067
The minimum iCa concentration (mmol/L)	total	0.94 ± 0.015	0.85 ± 0.026	0.0070
	prepartum	1.1 ± 0.015	1.02 ± 0.016	0.0009
	postpartum	0.94 ± 0.016	0.85 ± 0.026	0.0064
The number of days in hypocalcemia, based on tCa concentrations	total	1(0–1)	2(1–4)	0.0374
	prepartum	1(0–1)	0(0–0)	0.3087
	postpartum	0(0–0)	2(1–4)	0.0460
The minimum tCa concentration (mmol/L)	total	2.01 ± 0.038	1.81 ± 0.064	0.0131
	prepartum	2.21 ± 0.037	2.17 ± 0.029	0.3935
	postpartum	2.01 ± 0.038	1.81 ± 0.064	0.0131

Values are presented as median (lower-upper quartiles) or mean ± SE.

**Table 7 animals-15-00959-t007:** Average milk yield in the treated and control groups during the first 30 days of lactation.

Milk Yield	Weeks After Calving						*p*-Value		
Sampling	1	2	3	4	5	SEM	Trt	Week	Trt × Week
Control	25.99	29.75	35.80	39.18	43.16	2.09	0.165	<0.001	0.084
Treated	28.07	34.34	41.08	43.28	46.26				
*p*-value (pairwise)	0.492	0.129	0.083	0.174	0.303				

**Table 8 animals-15-00959-t008:** Milk aluminum concentrations in the treated and control groups.

Day 2				Day 7			
Treated		Control		Treated		Control	
Animal No.	Al (mg/L)	Animal No.	Al (mg/L)	Animal No.	Al (mg/L)	Animal No.	Al (mg/L)
2	<0.1 *	48	NA	2	<0.1	48	0.22
23	<0.1	108	<0.1	23	<0.1	108	<0.1
157	0.22	129	0.1	157	<0.1	129	0.15
181	<0.1	154	<0.1	181	<0.1	154	<0.1
5014	<0.1	156	<0.1	5014	<0.1	156	<0.1
5147	<0.1	178	0.16	5147	<0.1	178	<0.1
5280	<0.1	220	0.34	5280	NA	220	0.28
5438	0.34	5033	<0.1	5438	<0.1	5033	<0.1
5838	<0.1	5166	<0.1	5838	<0.1	5166	0.54
6151	<0.1	5629	<0.1	6151	0.35	5629	<0.1
6539	<0.1	5669	<0.1	6539	<0.1	5669	<0.1
7084	<0.1	6949	<0.1	7084	0.16	6949	<0.1
7251	0.14	7800	<0.1	7251	0.16	7800	<0.1
8164	<0.1	8083	0.28	8164	0.35	8083	<0.1
8283	0.15	8207	0.14	8283	<0.1	8207	<0.1
8373	<0.1	8384	<0.1	8373	<0.1	8384	<0.1
8597	<0.1	8440	<0.1	8597	<0.1	8440	<0.1

* The detection limit of the method was 0.1 mg/L. Values below the detection limit are displayed as <0.1 mg/L. NA = no data available.

## Data Availability

The data presented in this study are available upon request from the corresponding author.

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
