# Peer review of "Effects of a Phosphorus-Binding Feed Supplement on the Blood P and Ca Levels in Dairy Cows"

_animals, 2025, doi:10.3390/ani15070959_

Round 1

Reviewer 1 Report

Comments and Suggestions for Authors

Abstract:

L37. Put the keywords in alphabetical order.

Introduction

L41. Suboptimal means “less than the best possible” or “not the most effective”. I don’t think this word is suitable to characterize hypocalcemia.

L42. Hypocalcemia occurs more frequently during postpartum but can start prepartum too. Better use the transition period and not postpartum. On line 48 you stayed around calving, thus you said it can happen before calving

L46. Milk fever is the common name for hypocalcemia. So, there is a redundancy saying that hypocalcemia can lead to milk fever. I think you want to mean that leads to weakness and death.

L50. Diagnosis of hypocalcemia can be clinical too. However, for subclinical hypocalcemia, blood levels must be tested. Rephrase this statement.

L57. For milk production. Not of.

L60. And renal reabsorption too.

L62. The nutritional strategy is to prevent hypocalcemia, not prolonged hypocalcemia. We don’t ant animals to have clinical or subclinical hypocalcemia. Rephrase the sentence.

L.63. Oral calcium supplementation is not prevention, is treatment. And this is not a nutritional strategy. Nutritional strategy is low phosphorus forages, DCAD diets, aluminum silicate diet inclusion, etc.

Material and methods

  1. 94. Did you have committee approval to run an animal trial? Add this statement.

L96. What do you mean by thermoneutral? In dairy cows is between 5 and 25. However, I do think there were lower temperatures. Don’t think this is the best term. Maybe state the minimum and maximum temperatures.

L98. Did you check body condition score? Cows with high BCS are more prone to SCH and CH occurrence.

L108. Did not understand this statement. Diet was calculated to have refusals or not?

Table 1. Include a column with statistical analysis so there is more robustness on your group comparison.

L160. Did you block parity ( 2 or more), leghth of lactation, etc?

Results

The results are all put together and this is confusing. Not having a difference is also a result. Please separate them according to what was measured. Blood, milk, disease, etc. There are a lot of tables and graphs that could be more used. It seems that this section was left away.

In hypocalcemia is important to do de Delta calcium. i.e., the difference between times and what happens during pre and post partum. It is important also to observe the magnitude of the drop, the persistency of the drop and the time for recovery. Include this in your results.

Table 6. Did you test milk composition? The best comparison is to do energy-corrected milk.

Figures 1 and 2. I understand that you wanted to say 0.5 in the x-axis is because it was collected hours after parturition, however, is still day 0. Putting 0.5 just gets it confusing. What does the asterisk mean? Put in the figure descriptions. Also add when the baseline was taken.

Magnesium is part of calcium dynamics. However, you never mentioned it during the paper introduction but presented a figure in the results. Add magnesium importance to the introduction.

Figure 6. Do not add the p-value in the figure. Put the asterisk and add this information to the results section

Most figures and tables are not mentioned in the text! They must be!

Discussion.

Add the main findings and most importantly to the beginning of the discussion section.

I’ve missed the discussion about Ica and TCa. This was the main effect measured, and its discussion is shallow.

L 256. What is the problem with the low P levels? Hypophosphatemia has similar signs to hypocalcemia and can be easily mixed up.

L 308. Numerically lower doesn’t mean anything. If there wasn’t any statistical difference you cannot discuss the point of being numerically lower. Also, BHB was measured seven days apart. Ketosis occurs days after parturition, not immediately after. It is important to consider the body condition score to understand the animal's metabolism and this was not evaluated.

L301. If the cows consumed the whole amount of the TMR, with no leftovers, they were still hungry, and the amount provided wasn’t sufficient.

L319. Feed intake wasn’t measured, as well as there is no record of what happened with the body condition score. You cannot assume that.

I miss the comparison with other Ca bindings. Is AL-sulfate suitable for preventing hypocalcemia? Do the P levels drop as it happens to other bindings? Add this to the discussion.

Conclusion

I would remove the result that you did not detect significant amount of aluminion. The food safety part is already conclusion.

I miss in the  conclusion, is Al-sulfate suitable for preventing SCH and CH in dairy cows? Should we use it?

Author Response

AU: Thank you for your valuable comments and suggestions. We tried our best to improve the clarity and overall quality of the manuscript. All the changes we made in the text are indicated in yellow.

Abstract:
Rev.: L37. Put the keywords in alphabetical order.
AU: It is corrected.

Introduction
Rev.: L41. Suboptimal means "less than the best possible" or "not the most effective". I don't think this word is suitable to characterize hypocalcemia.
AU: The sentence is corrected

Rev.: L42. Hypocalcemia occurs more frequently during postpartum but can start prepartum too. Better use the transition period and not postpartum. On line 48 you stayed around calving, thus you said it can happen before calving
AU: Both sentences are corrected.

Rev: L46. Milk fever is the common name for hypocalcemia. So, there is a redundancy saying that hypocalcemia can lead to milk fever. I think you want to mean that leads to weakness and death.
AU: Yes, you are right. The sentence is corrected.

Rev.: L50. Diagnosis of hypocalcemia can be clinical too. However, for subclinical hypocalcemia, blood levels must be tested. Rephrase this statement.
AU: The sentence is rephrased.

Rev.: L57. For milk production. Not of.
AU: Corrected.

Rev.: L60. And renal reabsorption too.
AU: The sentence is augmented.

Rev.: L62. The nutritional strategy is to prevent hypocalcemia, not prolonged hypocalcemia. We don't ant animals to have clinical or subclinical hypocalcemia. Rephrase the sentence.
AU: The word' prolonged' is deleted.

Rev.: L.63. Oral calcium supplementation is not prevention, is treatment. And this is not a nutritional strategy. Nutritional strategy is low phosphorus forages, DCAD diets, aluminum silicate diet inclusion, etc.
Au: The sentences are rephrased.

Material and methods
Rev.: L94. Did you have committee approval to run an animal trial? Add this statement.
AU: Yes, we have approval, and it is mentioned in the Institutional Review Board Statement section of the manuscript, as it is written in the Authors' guidelines. We can add the statement here, too.

Rev.: L96. What do you mean by thermoneutral? In dairy cows is between 5 and 25. However, I do think there were lower temperatures. Don't think this is the best term. Maybe state the minimum and maximum temperatures.
AU:' thermoneutral conditions' deleted

Rev.: L98. Did you check body condition score? Cows with high BCS are more prone to SCH and CH occurrence. 
AU: The manuscript is now supplemented with baseline and end-of-study body condition scores and between-group comparisons.

Rev.: L108. Did not understand this statement. Diet was calculated to have refusals or not?
AU: The sentences in part L108-113 are indeed difficult to understand and inaccurate. The incorrect description also led to the other reviewer's comment regarding L301. In all experiments, TMR was mixed and fed to provide at least 5% refusal to ensure ad libitum feeding.
In the manuscript, we have corrected the mentioned part between as follows:
"......Ca balance (Tables 2 and 3). The daily amount of TMR in both groups was calculated to allow a 5% refusal to ensure ad libitum feeding. The cows of the treated group were given Al-sulfate at a dose of 400 g/day/cow until calving. As the TMR was offered ad libitum sufficient to allow at least 5% refusal, the measured amount of Al-sulfate in the mixer wagon was increased by 5%. The TMR was delivered once a day (between 0800 and 0900 h). Feed was pushed up three times per day. After calving, cows were moved to the fresh milking group and housed in a free-stall barn with straw bedding and ad libitum water supply. The composition of the TMR changed adaptively (Tables 2 and 3). The ration was adjusted daily to achieve a minimum 5% refusal and fed in two deliveries (0700 and 1700). TMR offered did not differ between the CTRL and TRT groups."

Rev.: Table 1. Include a column with statistical analysis so there is more robustness on your group comparison. 
AU: A column with the relevant p-values is now included in Table 1. 

Rev.: L160. Did you block parity ( 2 or more), length of lactation, etc?
AU: We did not consider parity and length of previous lactation as block effects since these factors are also not considered during farm management practices. All cows were fed and managed similarly, irrespective of parity and length of previous lactation. The potential confounding effect of such factors was handled by matching the control and experimental groups. Nevertheless, since the distribution of parity numbers and length of lactation days represent population values, the setting can also be considered stratified sampling. We are confident that such circumstances support the generalizability of our conclusions.

Results
Rev.: The results are all put together and this is confusing. Not having a difference is also a result. Please separate them according to what was measured. Blood, milk, disease, etc. There are a lot of tables and graphs that could be more used. It seems that this section was left away.
AU: We tried to reorganize this part. Also, non-significant differences are mentioned.

Rev.: In hypocalcemia is important to do de Delta calcium. i.e., the difference between times and what happens during pre and post partum. It is important also to observe the magnitude of the drop, the persistency of the drop and the time for recovery. Include this in your results.
AU: We are grateful that you have pointed this out. We have included comparisons of the number of hypocalcaemia days and minimum ionized and total Ca concentration between groups for the total study period, prepartum period and postpartum periods, respectively. A new table with the calculations you requested was created.

Rev.: Table 6. Did you test milk composition? The best comparison is to do energy-corrected milk.
AU: Indeed, it is the best comparison, but we could not measure milk composition.

Rev.: Figures 1 and 2. I understand that you wanted to say 0.5 in the x-axis is because it was collected hours after parturition, however, is still day 0. Putting 0.5 just gets it confusing. What does the asterisk mean? Put in the figure descriptions. Also add when the baseline was taken. 
AU: You are indeed right that 0.5 is misleading. We have corrected it to 0. The baseline was not indicated as it was assessed based on their expected calving date. However, the actual calving date could have slightly modified the time between calving and baseline measurements. Most times, the baseline was taken 14 days before the expected calving.

Rev.: Magnesium is part of calcium dynamics. However, you never mentioned it during the paper introduction but presented a figure in the results. Add magnesium importance to the introduction.
AU: The importance of Mg in Ca homeostasis is now mentioned.

Rev.: Figure 6. Do not add the p-value in the figure. Put the asterisk and add this information to the results section
AU: The p-values indicated tendencies. We do not wish to indicate tendencies, so we have removed them from the graph.

Rev.: Most figures and tables are not mentioned in the text! They must be!
AU: We double-checked the manuscript, and all tables and figures are referenced in the text. 

Discussion
Rev.: Add the main findings and most importantly to the beginning of the discussion section.
AU: This is like trying to overemphasize the good results, which are also described in the Results section, mentioned in the discussion, and even the main result is highlighted in the Conclusions section. We do not think that so many repetitions are necessary. We prefer to leave this part unchanged.

Rev.: I've missed the discussion about Ica and TCa. This was the main effect measured, and its discussion is shallow.
AU: Thank you, we added some sentences about this.

Rev.: L 256. What is the problem with the low P levels? Hypophosphatemia has similar signs to hypocalcemia and can be easily mixed up.
AU: Yes, indeed. However, there were no clinical cases during our study.

Rev.: L 308. Numerically lower doesn't mean anything. If there wasn't any statistical difference you cannot discuss the point of being numerically lower. Also, BHB was measured seven days apart. Ketosis occurs days after parturition, not immediately after. It is important to consider the body condition score to understand the animal's metabolism and this was not evaluated.
AU: Subclinical ketosis is highly frequent in dairy cows after calving, with the highest prevalence between 1 and 10 days postpartum, as we mention in the manuscript. We measured BHB levels during the entire study, from the enrollment until day 14 postpartum. We provided data about the BCS, as it was measured at the enrollment and on day 14. There was no difference between the two groups.

Rev.: L301. If the cows consumed the whole amount of the TMR, with no leftovers, they were still hungry, and the amount provided wasn't sufficient.
AU: The amount of TMR fed in the experiment was clarified in the revised manuscript. In the previous part of our response (see question L108), we explained how we corrected the sentences in the manuscript concerning feed intake for better interpretation. The daily amount of TMR in the experiment was calculated to allow a 5% refusal to ensure ad libitum feeding.

Rev.: L319. Feed intake wasn't measured, as well as there is no record of what happened with the body condition score. You cannot assume that.
AU: This is correct, the sentence is indeed an assumption. The sentence has been deleted.

Rev.: I miss the comparison with other Ca bindings. Is AL-sulfate suitable for preventing hypocalcemia? Do the P levels drop as it happens to other bindings? Add this to the discussion.
AU: We discussed this at the beginning of the discussion section. 

Conclusion
Rev.: I would remove the result that you did not detect significant amount of aluminion. The food safety part is already conclusion.
AU: The sentence is removed.

Rev: I miss in the  conclusion, is Al-sulfate suitable for preventing SCH and CH in dairy cows? Should we use it?
AU: Since there was no clinical milk fever during the study, we modified the conclusion that Al-sulfate is „Therefore, suitable for preventing subclinical hypocalcemia in dairy cows." 

Reviewer 2 Report

Comments and Suggestions for Authors
  1. Write a brief description of subclinical hypocalcemia and clinical hypocalcemia in line 42.
  2. Rewrite lines 65-71 for better understanding.
  3. Explain why the reduction in the phosphorous content has been effective in lines 72-73.
  4. In line 84, please mention who approved these additives and in which species.
  5. Add a reference for line 86.
  6. In line 95, what are the average ages and weights of the cows, and the offspring used in the study.
  7. Why did you all choose the months of February to April for the study and include the temperature instead of saying thermoneutral conditions in line 96.
  8. Why did you all choose dry cows for the study as most of your introduction is related to cows’ post-partum?
  9. Expand TMR in line 104.
  10. Rewrite lines 109-113 for clear understanding.
  11. Did the animal ethics committee approve your study, please mention the protocol number of the study.
  12. You have mentioned twelve occasions in line 131, but it does not match with the days you have mentioned in lines 132-133, please verify.
  13. Could you please explain why you have used caudal vessels instead of jugular veins to collect blood samples?
  14. In line 174, please explain which treatments have the values of 0.22, 0.18, and 0.14.
  15. Most of your introduction is related to Phosphorus and Calcium concentrations, but you have measured Mg and BHB concentrations, it would be good to add some background information related to Mg and BHB as well.
  16. In the table, you have included only seventeen animal data, I would like it if you could include all the animal data.
  17. Why did you choose 400g of Al sulfate as your treatment, why not lower or higher?
  18. How is Al-sulfate supplementation different from giving calcium supplementation before calving? Is Al-sulfate economical compared to Oral or IV calcium supplementation. Do you think you will have a difference in body weights.

Author Response

AU: Thank you for your valuable comments and suggestions. We tried our best to improve the clarity and overall quality of the manuscript. All the changes we made in the text are indicated in yellow.

Rev.: Write a brief description of subclinical hypocalcemia and clinical hypocalcemia in line 42.
AU: Lines 41-56 are about CH and SCH. However, we made some modifications to improve clarity.

Rev.: Rewrite lines 65-71 for better understanding.
AU: It is modified.

Rev.: Explain why the reduction in the phosphorous content has been effective in lines 72-73.
AU: According to the studies cited, it prevented hypocalcemia. The discussion section discusses a possible physiological background.

Rev.: In line 84, please mention who approved these additives and in which species.
AU: These are common food additives in human nutrition, and the reference is mentioned in the text (EFSA. Scientific opinion of the European Food Safety Authority (EFSA) Panel on Food Additives, Flavorings, Processing Aids and Food Contact Materials (AFC). Safety of aluminum from dietary intake. EFSA J 2008, 754, 1–34. https://doi.org/10.2903/j.efsa.2008.754)

Rev.: Add a reference for line 86.
AU: We added the reference.

Rev.: In line 95, what are the average ages and weights of the cows, and the offspring used in the study.
AU: The number of animals on the farm (600 cows + their offspring) is mentioned only to give the reader an idea of the location of the experiment. This is a large farm, and 600 cows are sufficient to select the focal animals (17 in the experimental and 17 in the control group). Not all 600 animals were included in the study, and especially not the offspring. We do not have data on the parameters of all the animals on the farm. The data of the cows enrolled in the study is displayed in Table 1.

Rev.: Why did you all choose the months of February to April for the study and include the temperature instead of saying thermoneutral conditions in line 96.
AU: We deleted the word' thermoneutral'. We chose these months to avoid summer heat stress, which could affect feed intake and animal health.

Rev.: Why did you all choose dry cows for the study as most of your introduction is related to cows' post-partum?
AU: Hypocalcemia is a problem occurring mainly in the postpartum period. However, nutritional prevention strategies aim to modify the Ca availability before calving to prepare the body for fast Ca mobilization after calving. Therefore, choosing dry cows is a logical step to test a nutritional prevention strategy.

Rev.: Expand TMR in line 104.
AU: The abbreviation has been resolved.

Rev.: Rewrite lines 109-113 for clear understanding.
AU: The sentences in part L108-113 are difficult to understand and inaccurate. In all experiments, TMR was mixed and fed to provide at least 5% refusal to ensure ad libitum feeding.
In the manuscript, we have corrected the mentioned part as follows:
"......Ca balance (Tables 2 and 3). The daily amount of TMR in both groups was calculated to allow a 5% refusal to ensure ad libitum feeding. The cows of the treated group were given Al-sulfate at a dose of 400 g/day/cow until calving. As the TMR was offered ad libitum sufficient to allow at least 5% refusal, the measured amount of Al-sulfate in the mixer wagon was increased by 5%. The TMR was delivered once a day (between 0800 and 0900 h). Feed was pushed up three times per day. After calving, cows were moved to the fresh milking group and housed in a free-stall barn with straw bedding and ad libitum water supply. The composition of the TMR changed adaptively (Tables 2 and 3). The ration was adjusted daily to achieve a minimum 5% refusal and fed in two deliveries (0700 and 1700). TMR offered did not differ between the CTRL and TRT groups."

Rev.: Did the animal ethics committee approve your study, please mention the protocol number of the study.
AU: Yes, and according to the Authors' guidelines, this should be mentioned in the Institutional Review Board Statement section of the manuscript. Of course, we can mention it here, too.

Rev.: You have mentioned twelve occasions in line 131, but it does not match with the days you have mentioned in lines 132-133, please verify.
AU: It is corrected.

Rev.: Could you please explain why you have used caudal vessels instead of jugular veins to collect blood samples?
AU: Taking blood samples from the caudal vein is an easy and accepted method. We wanted to avoid the stress of the cows being restrained during the sampling from the jugular vein.

Rev.: In line 174, please explain which treatments have the values of 0.22, 0.18, and 0.14.
AU: On average, the serum tCa concentration of TRT cows was 0.22, 0.18, and 0.14 mmol/L higher than control cows.

Rev.: Most of your introduction is related to Phosphorus and Calcium concentrations, but you have measured Mg and BHB concentrations, it would be good to add some background information related to Mg and BHB as well.
AU: We added some information to the text.

Rev.: In the table, you have included only seventeen animal data, I would like it if you could include all the animal data.
AU: If you mean Table 1, they are all there. We had 34 cows in total, 17 in the treated and 17 in the control group. As it is written in section 2.1.:' In total, 34 cows were assigned to the Control (CTRL, n=17) and Treatment (TRT, n=17) groups, respectively.

Rev.: Why did you choose 400g of Al sulfate as your treatment, why not lower or higher?
AU: This is a good point. On the one hand, based on previous literature data on Al-silicate, we tried to figure out what the effective dose might be for Al-sulfate. In addition, we conducted a smaller-scale dose-finding study (no data published), in which 400 g seemed to be effective.

Rev.: How is Al-sulfate supplementation different from giving calcium supplementation before calving? Is Al-sulfate economical compared to Oral or IV calcium supplementation. Do you think you will have a difference in body weights.
AU: We do not give Ca supplementation before calving at all. That would' suppress' parathormone and lead to hypocalcemia after calving. The nutritional strategy to prevent hypocalcemia during the prepartum period is to decrease the Ca absorbed from the intestines. These could be a low-Ca or a low-P diet, using anionic salts to decrease DCAD, Ca binders (Al-silicate), and P binders (Al-sulfate). So, Al-sulfate is a nutritional strategy used before calving. Therefore, it can not be compared with oral or i.v. Ca supplementation, which is the treatment of milk fever after calving.

Round 2

Reviewer 1 Report

Comments and Suggestions for Authors

Title:

Drop the “The”. Start with “effects of…”

Abstract

  1. 27: For aluminum silicate the minimum supplementation indicated is minimum 14 days, so probably the same for aluminum sulfate. In the material and methods section, you said that the cows were moved to close-up pens 3 weeks before calving. Thus, I think here, you were going to say the cows received the diet for a minimum of 14 days. Please clarify.

L33. Table 7 showed no milk production difference between treatments. Only a tendency on week 3. Saying that milk yield tended to be higher during the first 30 days is incorrect.

Introduction

Material and methods

Table 1. Drop the “?” in the BCS item.

Results

L205. Remove the ”.” after table 6.

Table 4> How can the SEM vakue for BHB be 14%?

Discussion

L348. Milk yield did not tend to be higher. The only difference was in week 3. This affirmation is trying to force a positive find for milk yield. I would drop this whole paragraph.

Author Response

AU: Thank you for your comments and suggestions. We indicated the modified text in yellow.

Title:

Rev.: Drop the “The”. Start with “effects of…”

AU: The title is corrected.

Abstract

Rev.: L27: For aluminum silicate the minimum supplementation indicated is minimum 14 days, so probably the same for aluminum sulfate. In the material and methods section, you said that the cows were moved to close-up pens 3 weeks before calving. Thus, I think here, you were going to say the cows received the diet for a minimum of 14 days. Please clarify.

AU: Yes, it is true. The cows received the treatment at least 14 days before calving. We clarified this in the text. We additionally modified the related text in the Materials and Methods section and added some explanation to the tables and figures indicating that -14 is the sampling time at enrolment.

Rev.: L33. Table 7 showed no milk production difference between treatments. Only a tendency on week 3. Saying that milk yield tended to be higher during the first 30 days is incorrect.

AU: We deleted this sentence from the abstract and modified the milk sections in the Results and Discussion sections.

Material and methods

Rev.: Table 1. Drop the “?” in the BCS item.

AU: Thank you. It is corrected. We also clarified whether the other data in Table 1 are displayed with SD or SE.

Results

Rev.: L205. Remove the ”.” after table 6.

AU: Unfortunately, I did not find this. If necessary, we can still correct this in the proof in the case of acceptance.

Rev.: Table 4: How can the SEM value for BHB be 14%?

AU: Thank you for pointing this out. The SEM displayed in % is indeed an unusual method. Due to the log-normal distribution of beta-hydroxy-butyrate values, we used logarithmic transformation before fitting models. However, values are hard for the reader to interpret, so retransformation to the original scale is advised. A confidence interval which is symmetrical on the log-scale (+/- 1.96*SE) becomes asymmetrical on the original scale due to the multiplicative nature of logarithmic models. Table 4 now contains the retransformed standard errors with the interpretation caveat in the caption.

Discussion

Rev.: L348. Milk yield did not tend to be higher. The only difference was in week 3. This affirmation is trying to force a positive find for milk yield. I would drop this whole paragraph.

AU: We kept the paragraph, since the negative results are also results, but we modified the sentences.

Reviewer 2 Report

Comments and Suggestions for Authors

Thanks for making the changes.

Author Response

Thank you for reviewing our manuscript.